# A Core and Valence-Level Spectroscopy Study of the Enhanced Reduction of CeO$_2$ by Iron Substitution—Implications for the Thermal Water-Splitting Reaction

Hicham Idriss 

Karlsruhe Institute of Technology (KIT), 76344 Eggenstein-Leopoldshafen, Germany; hicham.idriss@kit.edu

**Abstract:** The reduction of Ce cations in CeO$_2$ can be enhanced by their partial substitution with Fe cations. The enhanced reduction of Ce cations results in a considerable increase in the reaction rates for the thermal water-splitting reaction when compared to CeO$_2$ alone. This mixed oxide has a smaller crystallite size when compared to CeO$_2$, in addition to a smaller lattice size. In this work, two Fe-substituted Ce oxides are studied (Ce$_{0.95}$Fe$_{0.05}$O$_{2-\delta}$ and Ce$_{0.75}$Fe$_{0.25}$O$_{2-\delta}$; $\delta < 0.5$) by core and valence level spectroscopy in their as-prepared and Ar-ion-sputtered states. Ar ion sputtering substantially increases Ce4f lines at about 1.5 eV below the Fermi level. In addition, it is found that the XPS Ce5p/O2s ratio is sensitive to the degree of reduction, most likely due to a higher charge transfer from the oxygen to Ce ions upon reduction. Quantitatively, it is also found that XPS Ce3d of the fraction of Ce$^{3+}$ (u$_o$, u' and v$_o$, v') formed upon Ar ion sputtering and the ratio of Ce5p/O2s lines are higher for reduced Ce$_{0.95}$Fe$_{0.05}$O$_{2-\delta}$ than for reduced Ce$_{0.75}$Fe$_{0.25}$O$_{2-\delta}$. XPS Fe2p showed, however, no preferential increase for Fe$^{3+}$ reduction to Fe$^0$ with increasing time for both oxides. Since water splitting was higher on Ce$_{0.95}$Fe$_{0.05}$O$_{2-\delta}$ when compared to Ce$_{0.75}$Fe$_{0.25}$O$_{2-\delta}$, it is inferred that the reaction centers for the thermal water splitting to hydrogen are the reduced Ce cations and not the reduced Fe cations. These reduced Ce cations can be tracked by their XPS Ce5p/O2s ratio in addition to the common XPS Ce3d lines.

**Keywords:** cerium oxide reduction; cerium cation substitution; x-ray valence band spectroscopy; thermogravimetric analysis; XPS Ce4f; XPS Fe2p

## 1. Introduction

Cerium oxide is one of the most known stable redox oxides. This property makes it the oxide support of choice for automobile catalytic converters (three-way catalyst, TWC), which have been in use for decades [1]. The relative ease with which it goes through reduction and re-oxidation cycles while maintaining its crystallographic structure (fluorite) is the main reason for this [2]. This particular property also makes it among the best-known support oxides for other redox catalytic reactions, such as the water–gas shift reaction (WGSR) [3]. For both reactions (WGSR and TWC), the reduction of CeO$_2$ occurs chemically. In other words, the input of energy is from a hydrocarbon in the case of the TWC and from CO in the case of the WGSR [4]. This allows the redox cycle to occur at relatively low temperatures relevant for catalytic reactions.

CeO$_2$ is also one of the most active and stable known oxides for thermochemical water splitting to H$_2$ and O$_2$ for energy applications [5]. For this reaction, however, the reduction is considered in the absence of a reducing agent and where heat (from the sun) is the sole energy input. Temperatures typically above 1500 °C are needed for an appreciable reduction to take place [6]. This adds a considerable strain to the process, and so far, its possible applications would be very costly and largely unpractical. To this end, much work addressing the reduction of CeO$_2$ in order to understand its steps at the fundamental and applied levels has been pursued.

One of the strategies to enhance the reduction of $CeO_2$ is to mix it with other oxides. This may be grouped into three categories.

1. Size substitution (iso-valency): Compensation for lattice expansion. Upon the reduction of $Ce^{4+}$ cations to $Ce^{3+}$, the unit cell of $CeO_2$ increases. This is because the eight-coordinated $Ce^{4+}$ cation size is about 1 Å, while the eight-coordinated $Ce^{3+}$ cation size is about 1.1 Å. Substituting some of the $Ce^{4+}$ cations with metal cations with the same formal oxidation state ($M^{4+}$) but a smaller size compensates partly for the lattice expansion. This is particularly successful when $Zr^{4+}$ cations are used (size ca. 0.8 Å) [7,8]. While the substitution is valid up to about 50% (maintaining the fluorite structure of $CeO_2$) [9], phase segregation occurs at high temperatures (at 1000 °C or so) [10].

2. Charge transfer: substitution with oxidizable higher-valence cations. In this case, a fraction of $Ce^{4+}$ cations is substituted with metal cations that can donate electrons and themselves be oxidized [11]. The substitution of $Ce^{4+}$ with $U^{4+}$ was found to enhance the reduction of $CeO_2$, particularly at low levels [12,13]. Upon the removal of an oxygen atom, three $Ce^{3+}$ cations are formed (instead of two), and one $U^{4+}$ cation is oxidized to a $U^{5+}$ action. In addition, the fact that both oxides, $CeO_2$ and $UO_2$, have a fluorite structure and both cations have the same size makes them miscible for the entire ratio range [14]. The optimal dosing for the reduction of Ce cations is not clear yet, and neither is the temperature at which phase segregation occurs.

3. Charge compensation (alio-valencies): lattice distortion. While the substitution of $Ce^{4+}$ with metal cations of a lower oxidation state will create vacancies, these vacancies are not charged. In other words, there is no increase in electron charge. The effect is, however, clear; for example, the substitution of $Ce^{4+}$ cations with $Fe^{3+}$ cations (up to about 20%) results in a considerable reduction of the host oxide [15]. This is thought to be due to the distortion of the lattice structure, making it less stable and therefore enabling further reduction [16]. In recent work, this was found to be the case for high-temperature reduction (with no chemical input). Yet, considerable phase segregation occurred after one TCWS reaction cycle [17].

In the case of alio-valence substitution (the word substitution is sometimes mentioned as doping; strictly speaking, doping is the addition of another element in ppm or ppb amounts to change its electronic properties and is not necessary added as a substitution), a large number of elements were studied, and most showed that $Ce^{4+}$ reduction to $Ce^{3+}$ was enhanced. These include Co (4%) co-precipitation for the photoreduction of $CO_2$ to $CH_4$ [18], Co (23%) for COS hydrolysis [19], Co (20%) for Hg removal [20], Pr (10–50%), in which $Pr^{3+}$ increases the creation of oxygen vacancies and $Pr^{4+}$ increases the oxygen storage capacity [21] and Mn (12%), enhancing oxygen mobility via vacancy formation [22]. Computationally, a large body of work was also conducted to study the creation of oxygen vacancies upon alio-valent cation substitution. These include the following: Mn (DFT + U and HSE06) [23], Cu (DFT + U) [24], Ni (DFT + U and HSE06) [25] and (DFT + U) [26], and other rare earth elements (Sc, Y, Gd, La) by DFT + U and Monte Carlo simulations [27].

Focusing on Fe substitution, considerable work at the experimental and theoretical levels has been conducted, and a few studies are mentioned here. Fe substitution of less than 30% increased the number of oxygen vacancies [28] and the selective catalytic reduction of NO to $NH_3$. The DFT +U results of Fe-doped $CeO_2$ (111) indicated that the oxygen vacancy formation energy is lower when compared to ceria alone and that Fe tends to be the center of the oxygen vacancy clusters [29]. Improved CO conversion over a $CeO_2$-$Fe_2O_3$ mixed oxide was also seen and attributed to the formation of more mobile oxygen atoms in the redox cycle [30]. Fe substitution was previously studied for the thermochemical water-splitting reaction among other cation substitutions (Mn, Ni, and Cu), and the $CeO_2$ substitute was found to be still active after four reaction cycles ($O_2$ and $H_2$ release) [31] at 1273 K (1000 °C). In another work for the same reaction, Fe at 5% was tested at 1550 °C and was found to be more active (a higher production rate per unit weight) and faster (a higher rate of release of hydrogen) than $CeO_2$ alone. However, considerable phase segregation

occurred due to the very high thermal reduction, 1550 °C, after one cycle of reaction [17]. This work is an extension of a previous one, focusing on the changes that occurred upon the reduction of Ce cations in the presence of Fe cations. Since during the reduction of $Ce^{4+}$ cations to $Ce^{3+}$ cations, $Fe^{3+}$ cations would also be reduced to $Fe^{2+}$ and $Fe^0$, monitoring the changes that occur for both elements is needed. Also, as it will be shown, it seems that the ratio of the XPS Ce5p/O2s shallow core level lines is sensitive to the degree of reduction.

## 2. Results and Discussion

The prepared $CeO_2$ and Fe-substituted $CeO_2$ were studied in some detail previously by TEM, EELS, XRD and TPR, among other methods [17]. Figure 1 gives a brief description of $Ce_{0.95}Fe_{0.05}O_{2-\delta}$ and $Ce_{0.75}Fe_{0.25}O_{2-\delta}$. Both oxides are composed of small crystallites upon calcination (500 °C). These were about 8 and 5 nm in size for $Ce_{0.95}Fe_{0.05}O_{2-\delta}$ and $Ce_{0.75}Fe_{0.25}O_{2-\delta}$, respectively. The TEM images show that they are dominantly (111) terminated. These crystals are identical to those of $CeO_2$ [17], which is composed of crystallites of about 14 nm in size. The decrease in crystallite sizes upon Fe substitution is also similar to that previously reported by others [32]. This is most likely due to the change in ionicity of the sol–gel medium during the precipitation [33]. Also, as seen, the only observed phase (XRD, TEM) is the fluorite one. The unit cells show a slight decrease due to Fe substitution (0.543, 0.539 and 0.536 nm for $CeO_2$, $Ce_{0.95}Fe_{0.05}O_{2-\delta}$ and $Ce_{0.75}Fe_{0.25}O_{2-\delta}$, respectively). As shown in Figure 1C,D, up to 700 °C, there is a gradual, small increase in the lattice and crystallite sizes. An abrupt change occurs at 900 °C, where considerable sintering is seen and is accompanied by the appearance of lines due to Fe oxides, indicating the beginning of the phase segregation process, which was accentuated by 1100 °C.

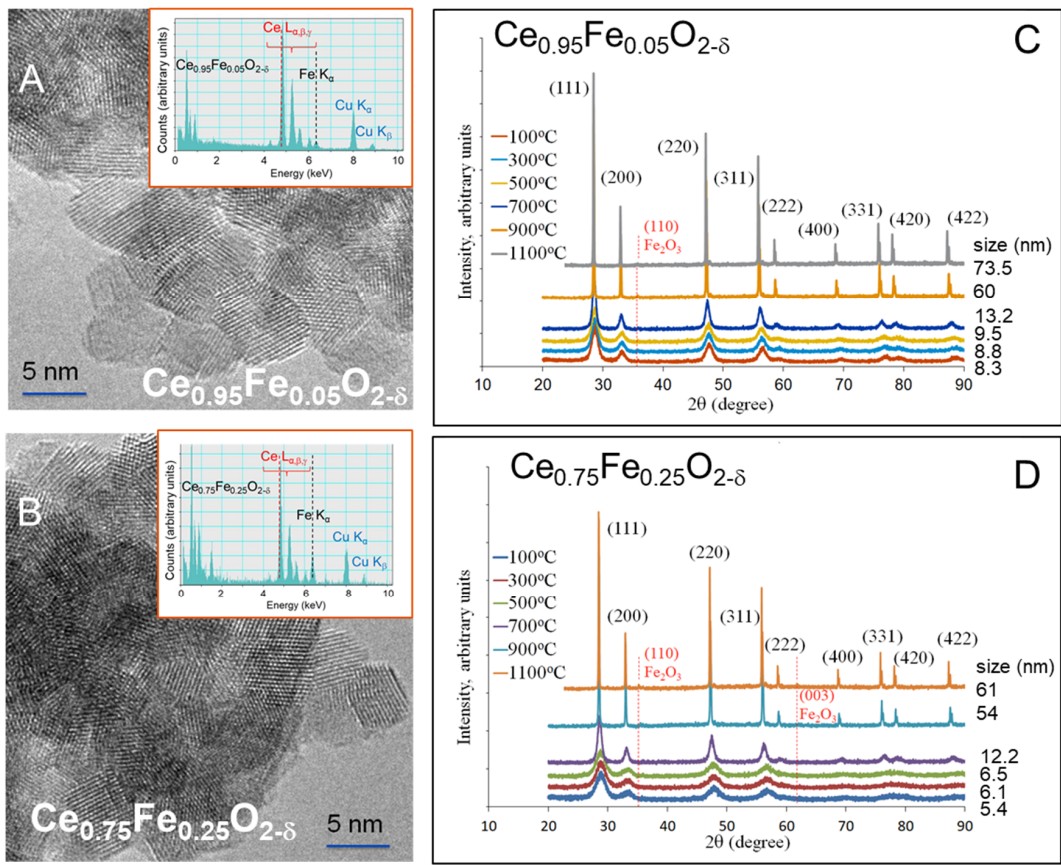

**Figure 1.** Transmission electron microscopy (TEM) (A and B), energy-dispersive x-ray (EDX) (insets in (**A**,**B**) and x-ray diffraction (XRD) of (**C**) $Ce_{0.95}Fe_{0.05}O_{2-\delta}$ and (**D**) $Ce_{0.75}Fe_{0.25}O_{2-\delta}$. The figures are adapted with permission from [17].

Figure 2A,B present the valence band and shallow core levels (Ce5p, O2s, and Ce5s) of $Ce_{0.75}Fe_{0.25}O_{2-\delta}$ and $Ce_{0.95}Fe_{0.05}O_{2-\delta}$ before and after 1, 2 and 5 min of argon ion sputtering. The reduction of the oxides was conducted by Ar ion sputtering. This method has advantages and disadvantages when compared to hydrogen reduction or thermal reduction. Thermal reduction of these oxides requires heating to at least 1500 °C. The mixed oxide at this temperature largely segregates, and therefore, the analysis would not be useful. Reduction using molecular hydrogen in UHV is not possible because of its very low sticking coefficient on oxides. It is possible to reduce oxides with atomic hydrogen, but this method was not available. Argon ion sputtering to reduce oxides relies on the mass difference between the metal cation and oxygen anions and the stability of the reduced phase thus created at given experimental conditions. $Ce^{3+}$, $Fe^{2+}$, and $Fe^{0}$ are stable in UHV conditions. In other words, once formed, not much change is seen within the investigation time (a few hours). The drawback of Ar ion sputtering is the loss of surface structure (the formation of an amorphous layer due to the high energy ion bombardment used, in this work, 1 kV).

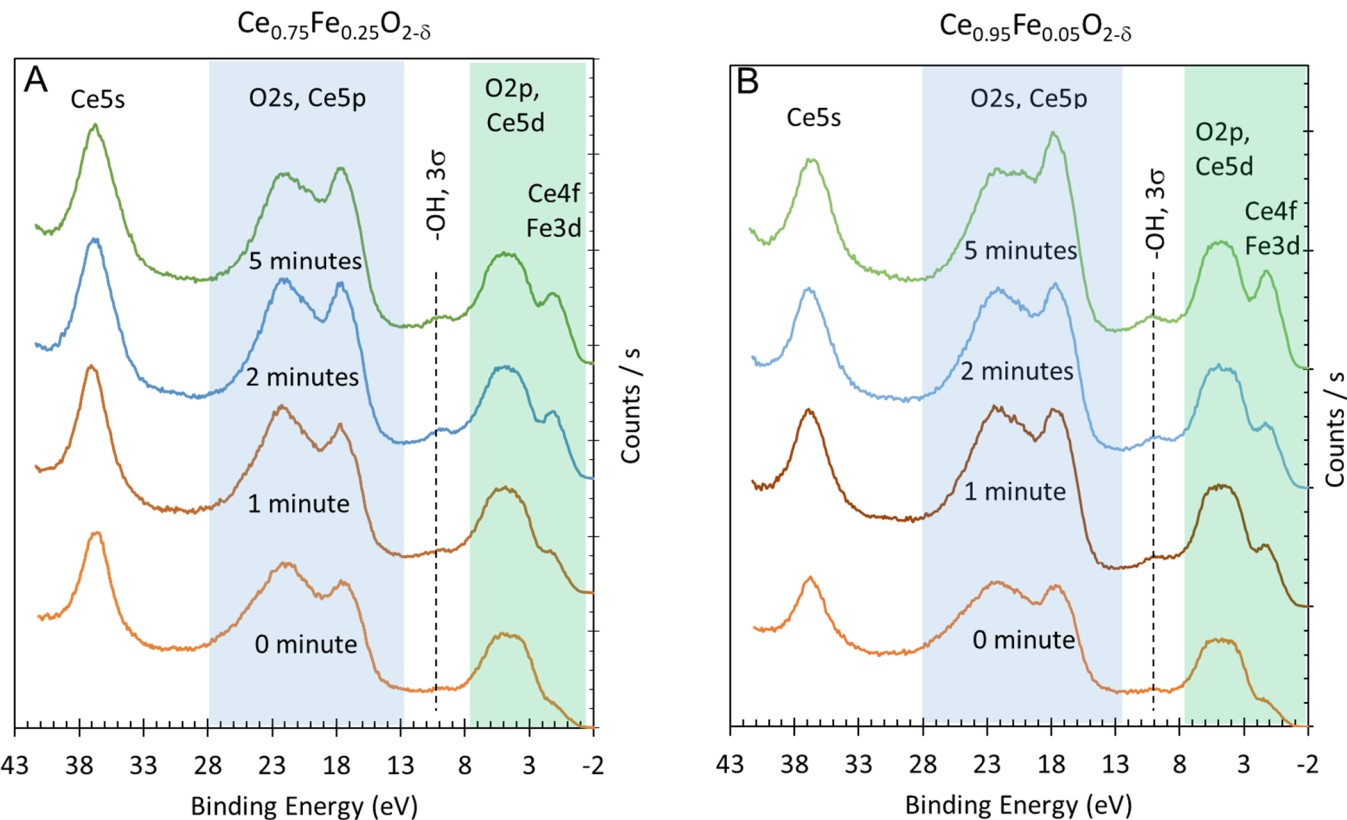

**Figure 2.** (**A**) Valence band XPS (green shaded) and Ce5p, O2s (blue-shaded) and Ce5s spectra of $Ce_{0.75}Fe_{0.25}O_{2-\delta}$ before (0 min) and after argon ion sputtering (x min). (**B**) Valence band XPS (green shaded) and Ce5p, O2s (blue-shaded) and Ce5s spectra of $Ce_{0.95}Fe_{0.05}O_{2-\delta}$ before (0 min) and after argon ion sputtering (x min).

The initial spectra of the as-prepared oxides (0 min) are dominated by the O2p, O2s, Ce5p and Ce5s lines and are hardly distinguishable for both oxides. There are traces of unavoidable surface hydroxyls (-OH, 3 σ) at a binding energy of ca. 10 eV and some contribution from reduced Ce cations (Ce4f, $Ce^{3+}$) and reduced Fe cations (Fe3d, oxidation state < +3), both below the O2p band with a binding energy of 0.5–2 eV. Upon argon ion sputtering, three changes are noticeable: (i) An increase in the signal before the O2p reductions (due to increased concentrations of $Ce^{3+}$ and $Fe^{+x}$, x < +3). (ii) An increase in surface hydroxyls. (iii) A relative increase in Ce5p signal with respect to the O2s signal. For

(i), the increase is expected and is treated in more detail in this work to extract quantitative information. For (ii), the increase has been seen before and is linked to the increase in the sticking coefficient of dissociatively adsorbed water (in the background) over a reduced metal oxide when compared to its stoichiometric form. The ion bombardment causes a reduction due to oxygen removal (as atoms), and the remaining electrons (two for each oxygen atom removed) are transferred to $Ce^{4+}$ (and $Fe^{3+}$) to reduce them. The creation of the oxygen vacancy leads to the preferential dissociative adsorption of a water molecule. This results in the formation of two pairs of surface hydroxyls for each oxygen vacancy healed. This is not the purpose of this study and will not be further treated. For (iii), this observation is unexpected and is treated here qualitatively.

Figure 3A,B present the valence band region, in which the signal below the O2p orbital was fitted by two peaks: the first at about 0.4 eV and the second at ca. 1.5 eV below the Fermi level. There are no noticeable changes in the large O2p lines' shapes with sputtering time. For both oxides, the signal attributed to the Ce4f orbital is higher than that of the Fe3d orbital. The increase of the XPS Ce4f signal upon reduction is more pronounced in the case of a low Fe concentration when compared to the other oxides. This is consistent with the other core levels (see below) as well as with the TCWS results (also see below). Computing the peak areas of the two peaks seems to indicate that there is no incentive to further reduce Ce cations with a higher % of Fe. At a low Fe %, increasing the reduction time affects mostly the Ce cations, while at a higher Fe %, it favors Fe reduction. This might be simply due to the probability of hitting the atoms during bombardment. At a high Fe %, the probability of oxygen removal adjacent to Fe atoms is high, and therefore more Fe is reduced, while at a low Fe %, the oxide is more homogeneous and the chemical effect on the reduction is higher.

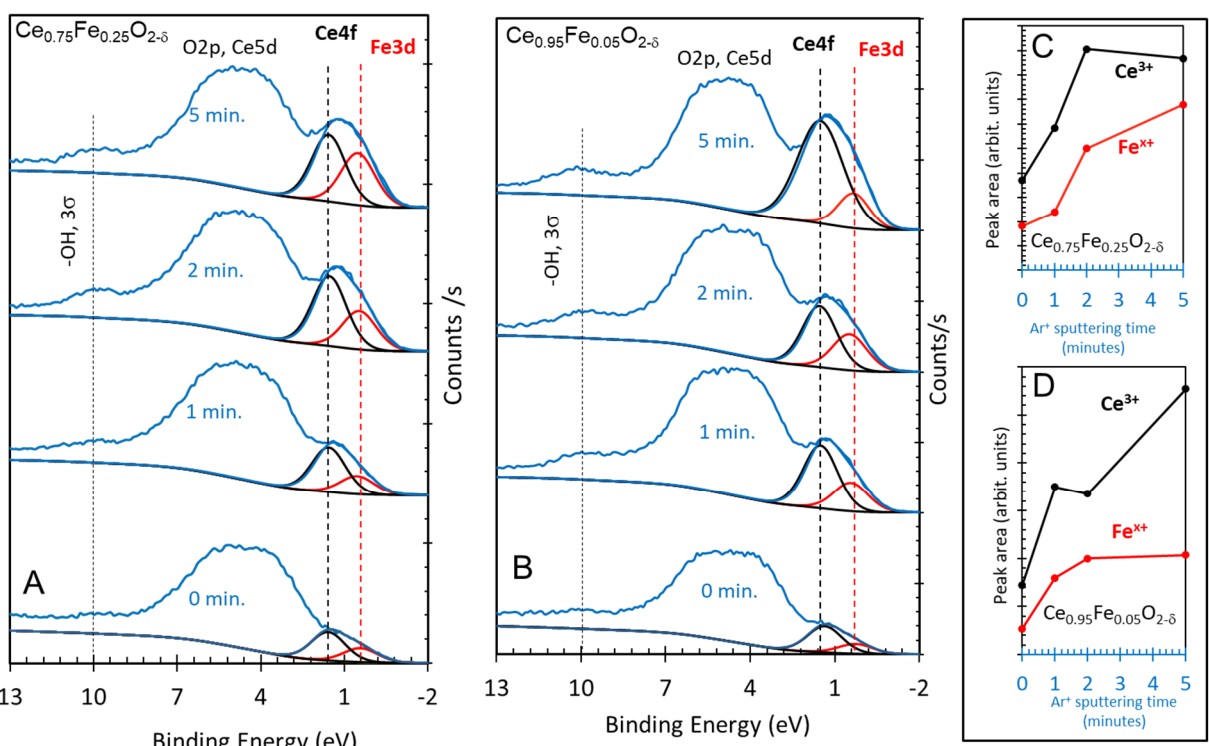

**Figure 3.** (**A**) Valence band XPS of as-prepared $Ce_{0.75}Fe_{0.25}O_{2-\delta}$ after 1, 2 and 5 min of Ar ion sputtering. (**B**) Valence band XPS of as-prepared $Ce_{0.95}Fe_{0.05}O_{2-\delta}$ after 1, 2 and 5 min of Ar ion sputtering. (**C**,**D**) Computed peak areas of fitted Ce4f (FWHM = 1.5 eV) and Fe3d (FWHM = 1.5 eV) signals.

Figure 4A,B present the valence band together with the Ce5p and O2s lines. The spectra are baseline-subtracted and then normalized to highlight the differences. The ion sputtering of both oxides results in a preferential increase in the Ce5p signal when compared to the

O2s lines. In addition, the O2s line becomes narrower upon ion sputtering. There is no noticeable shift in the binding energy before or after ion bombardment. Similar experiments were conducted on $CeO_2$, and no change was seen (SI Figure S1). Obtaining quantitative information from polycrystalline oxides is more difficult because of the unavoidable grain boundary effect (channeling) and shadowing, both of which would make ion sputtering less efficient when compared to sputtering thin films or single crystals. Nevertheless, some qualitative information may be drawn from comparisons to previous work conducted by others. One of the most relevant to this present work is a study on the ion bombardment [34] of $CeO_2$, in which it was shown (Figure 3a,b of ref. [34]) that the relative ratio is indeed increased in favor of the Ce5p orbital. The exact position of the O2s orbital with respect to the $Ce5p_{1/2}$ orbital is not clear. In this work, it is put after the $Ce5p_{1/2}$ lines in line with other work, although others have put it in between the $Ce5p_{3/2}$ and $Ce5p_{1/2}$ energy positions based on the relativistic computation of $CeO_8$ and $Ce_{63}O_{216}$ clusters [35]. One of the motivations of studies of the O2s and Ce5p lines is charge transfer, where these lines, because of their quasi-degenerate energy positions, are sensitive to the oxidation state of Ce cations. The spectra in Figure 4 are similar to those reported for a thin film of $CeO_2$ grown on Rh(111) [36] excited with photon energy equal to 125 eV (Ce4d-Ce4f resonance). The spectra are also similar to those of irradiated (with Xe ions) $CeO_2$ thin film and bulk [34]. The authors pointed out the final state effect ($3d^9 4f^1 OVMO^{-1}$ (outer valence molecular orbital, OVMO) and $3d^9 5p^5 np^1$ (inner valence molecular orbital, INVO)). In another work [33], the authors indicated the Ce5p atomic orbitals participate in the formation of both OVMO and IVMO, where a large part of the latter is taken by the filled $Ce5p_{1/2}$, $5p_{3/2}$ and O2s atomic shells.

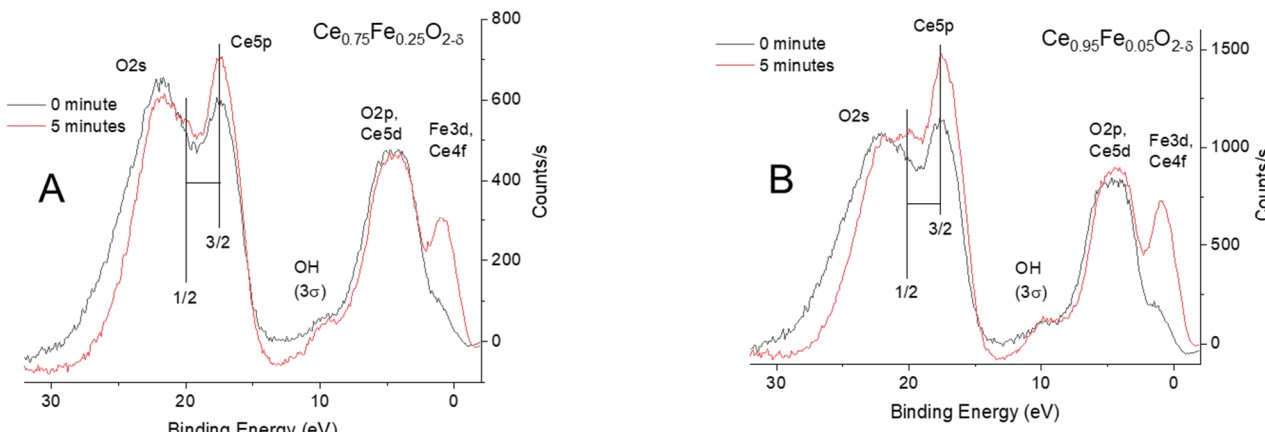

**Figure 4.** (**A**) Normalized valence band XPS and the Ce5p and O2s spectra of $Ce_{0.75}Fe_{0.25}O_{2-\delta}$ before (0 min) and after 5 min of argon ion sputtering. (**B**) Normalized valence band XPS and the Ce5p and O2s spectra of $Ce_{0.95}Fe_{0.05}O_{2-\delta}$ before (0 min) and after 5 min of argon ion sputtering. Note the change in intensity of the Ce5p signal when compared to the O2s signal.

The spectra in Figure 4 are fitted in Figure S2. Four peaks are considered ($Ce5p_{3/2,1/2}$ and two for O2s). The two O2s peaks are for the two different (bulk and surface oxygen atoms) environments and are separated by about 2.5 eV (similar to the O1s signal). It is to be noted that similar attribution was given for the O2s signal to clean and glycine-dosed surfaces, where the high-binding O2s line is due to surface hydroxyls and carboxylates [37].

Figure 5 presents the XPS Ce3d spectra of the Ce cations in the fresh and Ar-ion-reduced samples. The presence of both oxidation states of Ce cations is clear even in the freshly prepared oxides. However, in this case, the $Ce^{3+}$ cation concentration is small, and they most likely reside in deeper layers below the surface. In Figure 5 the lines' positions and attributions are shown ($3d_{5/2}$: u, u'' and u''', and $3d_{3/2}$: v, v'' and v''' for $Ce^{4+}$ cations, and $3d_{5/2}$: $u_o$ and u' and $3d_{3/2}$: $v_o$ and v' for $Ce^{3+}$ cations). Upon fitting these peaks, information can be obtained. In Figure 5 and the inset table, a comparison between the

fresh and the most reduced samples is made. The presence of Fe at a 5% concentration resulted in a more pronounced $Ce^{3+}$ concentration when compared to the presence of Fe at a 25% concentration. This is in line with the valence band results presented in Figures 2 and 3. $CeO_2$ alone showed a very mild increase upon reduction; this is also in line with the virtually no change in the Ce5p/O2s lines in Figure 1. Ar ion sputtering relies on the momentum transfer of incoming ions (in this case, 1 keV of kinetic energy), which results in breaking the chemical bonds. As such, there should be no difference between $CeO_2$ and Fe-substituted $CeO_2$ since the incoming ions have much more energy than the chemical bond, assuming complete energy transfer. However, this is a cascade reaction where the energy transfer occurs consecutively, and therefore the last steps of an incoming ion (before it leaves the material or is implanted in it irreversibly) would be more efficient for weaker bonding. This is particularly important for polycrystalline nanoparticles, where interparticle pore diffusion dominates.

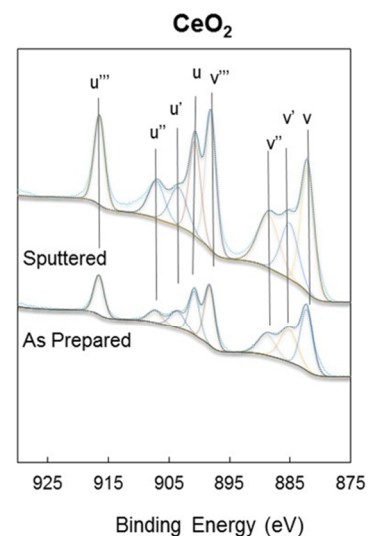

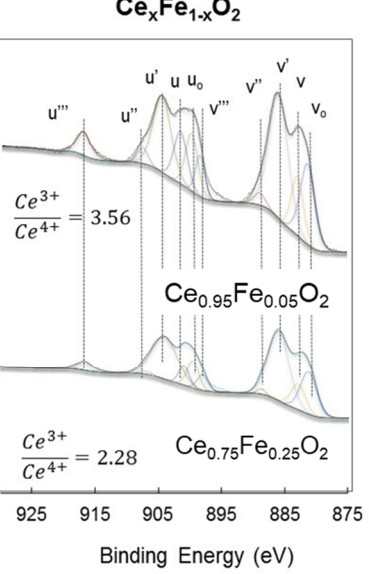

**Figure 5.** XPS Ce3d spectra of as-prepared $CeO_2$ (left panel), $Ce_{0.75}Fe_{0.25}O_{2-\delta}$ and $Ce_{0.95}Fe_{0.05}O_{2-\delta}$ (right panel) after 5 min of Ar ion sputtering. Also shown are the computed contributions of the $Ce^{3+}$ and $Ce^{4+}$ cations.

Based on the Ce4f/O2p, Ce5p/O2s and Ce3d (for $Ce^{4+}$ and $Ce^{3+}$) XPS signals, Table 1 is constructed to provide an estimate of the reduction of Ce cations in this work.

**Table 1.** Extracted quantitative values from XPS Ce3d, Ce4f, Ce5p and Fe2p spectra and thermochemical water splitting to hydrogen over reduced oxides. The XPS data are those for reduced oxides after 5 min of argon ion sputtering. The hydrogen production data are at 700 °C under $N_2$ from prior hydrogen-reduced oxides at 700 °C.

| Oxide | $Ce^{3+}/Ce^{4+}$ (Ce3d) | $[Ce4f + Fe3d^x]/O2p$ | $Fe^0/Fe^{3+}$ (Fe2p) | Ce5p/O2s | $H_2$ Production (mol/g) |
|---|---|---|---|---|---|
| $CeO_2$ | 0.2 | - | - | 0 | $0.2 \times 10^{-6}$ |
| $Ce_{0.75}Fe_{0.25}O_{2-\delta}$ | 2.3 | 0.3 | 0.5 | 1.6 | $7.4 \times 10^{-6}$ |
| $Ce_{0.95}Fe_{0.05}O_{2-\delta}$ | 3.6 | 0.4 | 0.6 | 1.75 | $11.4 \times 10^{-6}$ |

The XPS core levels of iron oxides are among the most studied for oxides [38]. There are three common oxidation states for the oxides: $Fe^{3+}$, such as in $Fe_2O_3$, $Fe^{2+}$, such as in

FeO, and $Fe_3O_4$, in addition to metallic Fe. The spectra are complicated by the presence of satellites [39–41], iron hydroxide (FeOOH) [42,43] and many multiplets [44]. The binding energy of the XPS Fe2p signal is about 707 eV for metallic iron, 710 eV for $Fe^{2+}$ and 711 eV for $Fe^{3+}$ cations. In the present study, a further complication arises from the presence of Ce Auger lines in the Fe2p region. Moreover, although charge neutralization was used, the as-prepared oxide always had wider peaks when compared to that reduced upon ion sputtering. Therefore, peak areas are to be taken with an estimate of 20% errors and binding energies within a 0.5 eV accuracy. Both oxides show very similar spectra and trends upon reduction. While, as expected, the as-prepared oxides contained $Fe^{3+}$ cations, they, however, also contained $Fe^{2+}$ and some metallic iron. Some small % of $Fe^{2+}$ might be formed during the preparation (Figure 6). The presence of metallic iron was not expected and is most likely due to interstitial atoms formed due to strong lattice distortion. The insets in both figures show the trend during the reduction. It was opted chose to subtract the $Fe^{2+}$ and $Fe^0$ contributions of the as-prepared oxides to see the trend. In both cases, the amount of $Fe^{2+}$ increases, similar to a previous study [45], then decreases to zero, indicating that within the reduction time studied, all reducible $Fe^{3+}$ cations were transformed to metallic Fe. In line with the valence band results, it appears that both $Ce^{4+}$ and $Fe^{3+}$ cations are more efficiently reduced when in small amounts ($Fe_{0.05}$), as seen in Figure 6B, although the difference is not dramatic.

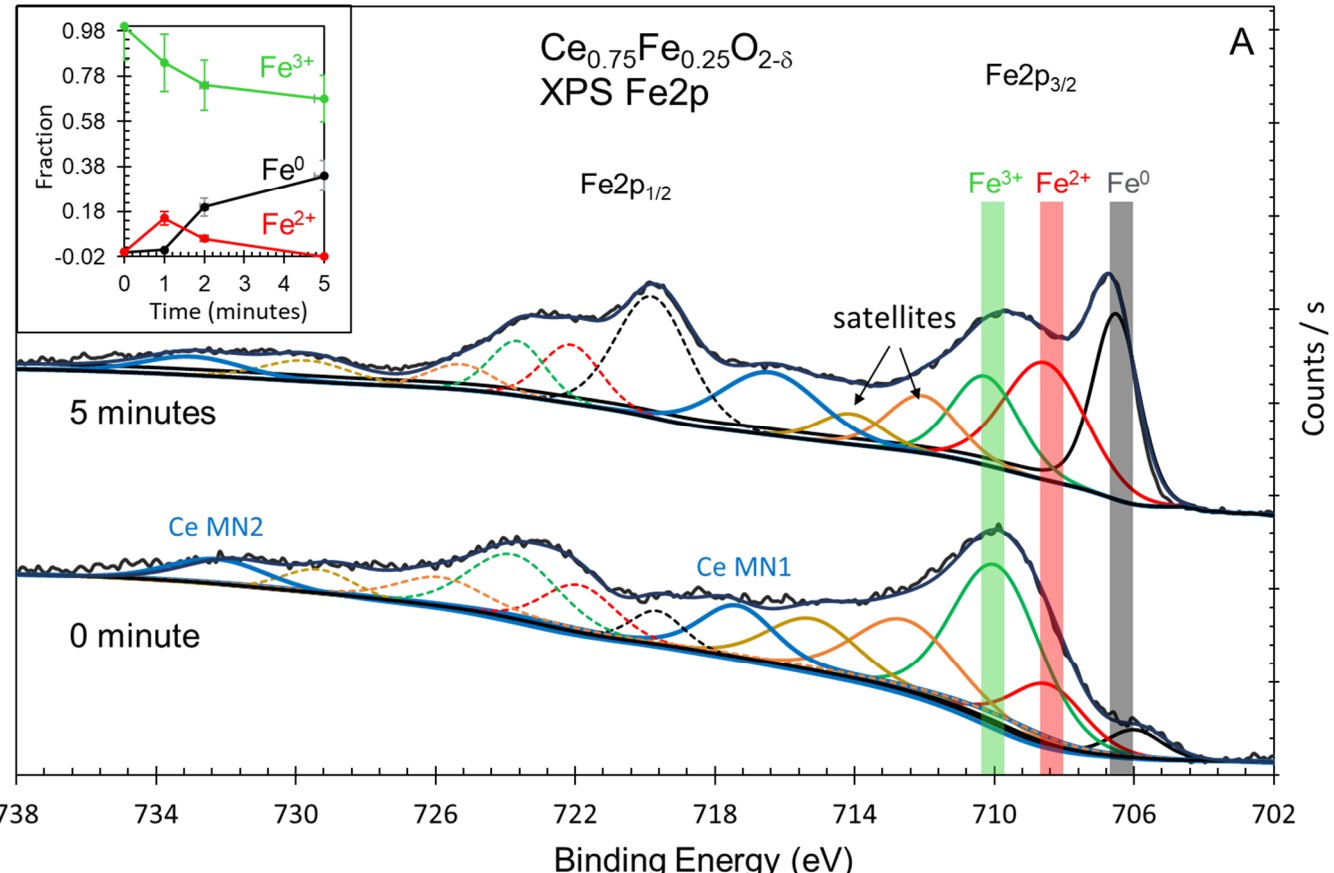

**Figure 6.** *Cont.*

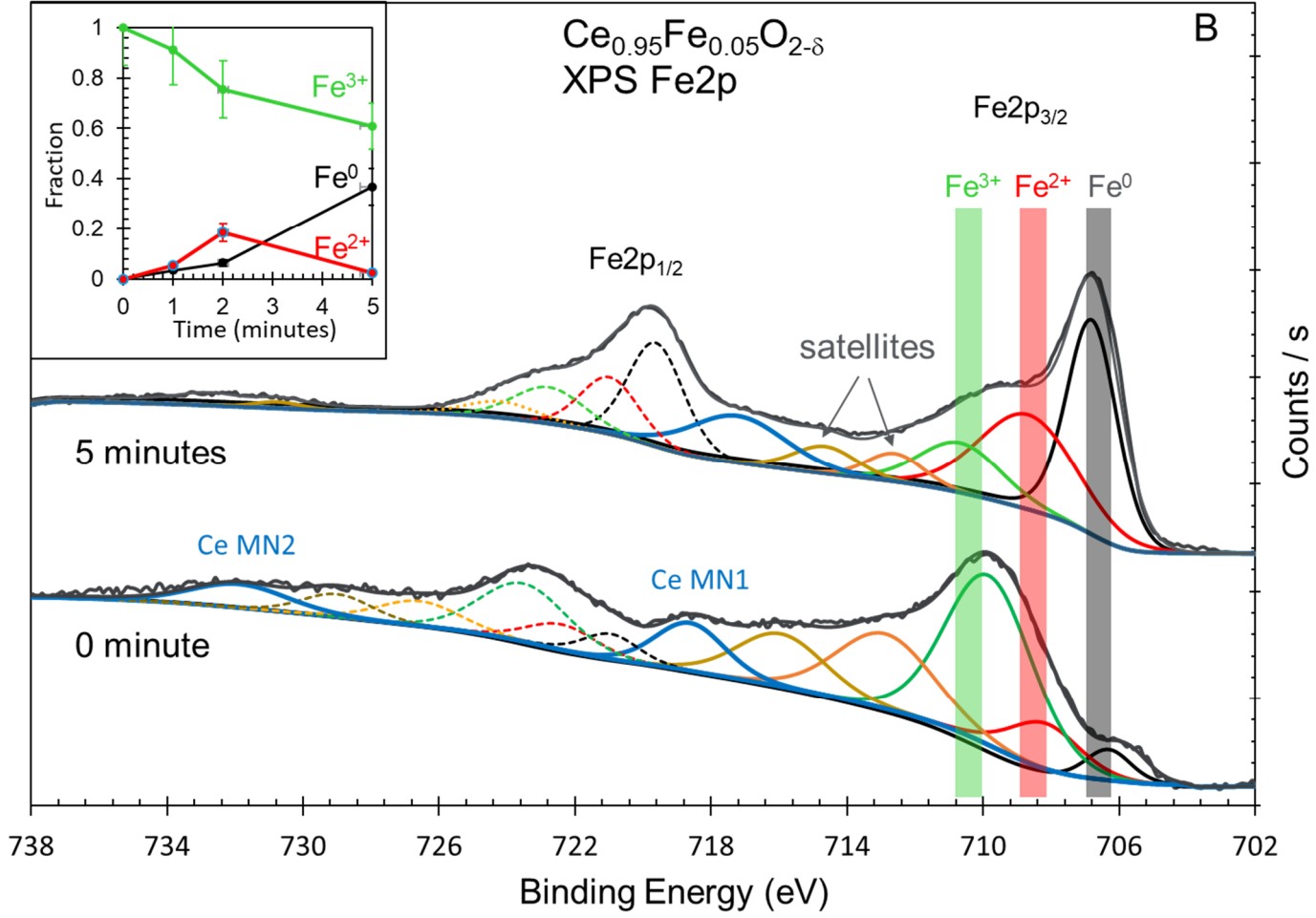

**Figure 6.** (**A**) XPS Fe2p spectra of as-prepared $Ce_{0.75}Fe_{0.25}O_{2-\delta}$ after 5 min of Ar ion sputtering. (**B**) XPS Fe2p spectra of as-prepared $Ce_{0.95}Fe_{0.05}O_{2-\delta}$ after 5 min of Ar ion sputtering. Insets in (**A,B**): quantitative analysis of XPS Fe2p spectra of $Ce_{0.75}Fe_{0.25}O_{2-\delta}$ and $Ce_{0.95}Fe_{0.05}O_{2-\delta}$ before and after sputtering at the indicated time.

Figure 7 presents the hydrogen production of the three oxides from water at 700 °C. The point of the figure is to relate the production to the observed reduced states of the oxides. Because, as indicated, the two mixed oxides are not segregated when heated to this temperature, a link to the spectroscopic measurements would be relevant even though, as such, it is not practical (since the oxides were previously reduced with molecular hydrogen). $CeO_2$ alone shows negligible activity, while the Fe-substituted cerium oxides are active (the production is given in Table 1). The activity of the $Ce_{0.95}Fe_{0.05}O_{2-\delta}$ catalyst is almost twice that of the $Ce_{0.75}Fe_{0.25}O_{2-\delta}$ catalyst. This suggests that the activity is more linked to $Ce^{3+}$ than to $Fe^0$ (based on the XPS Ce3d and XPS Ce5p/O2s spectra). This is because the content of $Fe^0$ in both reduced oxides was similar (XPS Fe2p), while that of $Ce^{3+}$ was higher for $Ce_{0.95}Fe_{0.05}O_{2-\delta}$ than for $Ce_{0.75}Fe_{0.25}O_{2-\delta}$.

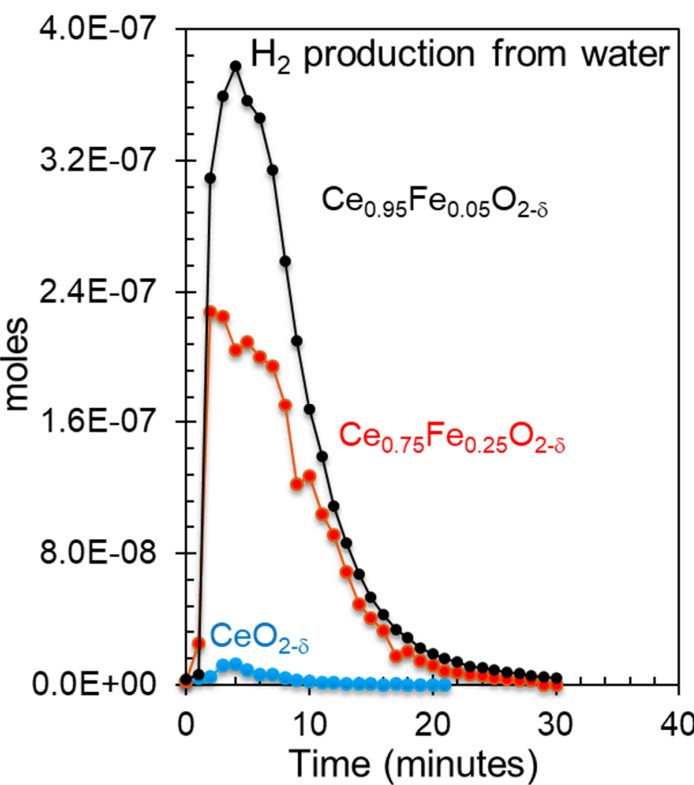

**Figure 7.** Hydrogen production from water over $CeO_2$, $Ce_{0.75}Fe_{0.25}O_{2-\delta}$ and $Ce_{0.95}Fe_{0.05}O_{2-\delta}$, which were previously reduced with hydrogen at 700 °C at one atmosphere.

## 3. Experimental Section

The $CeO_2$, $Ce_{0.95}Fe_{0.05}O_{2-\delta}$ and $Ce_{0.75}Fe_{0.25}O_{2-\delta}$ catalysts were synthesized by the co-precipitation method already presented in other work [17]. In brief, cerium (III) nitrate hexahydrate (from Sigma Aldrich, St. Louis, MO, USA) was dissolved in deionized water alone or together with the needed amount of iron (III) nitrate (from Sigma Aldrich). $NH_4OH$, used as a precipitating agent (70 vol. %), was added to the solution containing the metal cations while stirring until a pH > 9 was reached. After the filtration of the precipitate, it was washed using deionized water until a neutral pH was obtained. The precipitate (in the form of metal hydroxides) was then dried in an oven at 100 °C overnight, then crushed, loaded into a crucible and calcined in air at 500 °C for 12 h with a temperature ramp of 15 °C min$^{-1}$.

X-ray diffraction (XRD) data were collected using a PANalytical EMPYREAN diffractometer (Malvern, UK) in Bragg–Brentano geometry with Cu K$_\alpha$ excitation at 45 kV and 40 mA and a linear position-sensitive detector. The diffractometer was configured with a 0.25° diverging slit, a 0.5° anti-scattering slit, 2.3° Soller slits, and a Ni filter. Data were acquired in continuous scanning mode over the 2$\theta$ range 10–90° at a step interval of 0.01° and 0.5 s per step.

Transmission electron microscopy (TEM) studies were conducted using a Titan ST microscope (FEI company, Hillsboro, OR, USA). It operated at an accelerating voltage of 300 kV and was equipped with a field emission electron gun, a 4k×4k CCD camera, a Gatan imaging filter (GIF), and a Gatan microscopy suite (GMS).

X-ray photoelectron spectroscopy (XPS) was conducted using a Thermo scientific ESCALAB 250 Xi. Spectra (Waltham, MA, USA) were calibrated with respect to the C1s line at 284.7 eV. The Fe2p, O1s, Ce3d, Ce4d, C1s, O2s, Ce5s, Ce5p, and the valence band (O2p, Ce4f and Fe3d) binding energy regions were scanned. The typical acquisition conditions were as follows: pass energy = 20 eV and scan rate = 0.1 eV per 200 ms. Ar$^+$- bombardment was conducted with an EX06 ion gun at a 1 kV beam energy and a 10 mA emission current.

The sputtered area of $900 \times 900~\mu m^2$ was larger than the analyzed area of $600 \times 600~\mu m^2$. Self-supported oxide disks, approximately 0.5 cm in diameter, were loaded into the chamber. Data acquisition was carried out using the Avantage Data System V5 software. Charge neutralization was used for all the samples (1 eV).

The water-splitting experiments were conducted as follows: 0.25 g of the metal oxide ($CeO_2$, $Ce_{0.95}Fe_{0.05}O_{2-\delta}$ or $Ce_{0.75}Fe_{0.25}O_{2-\delta}$) was put in a tubular quartz reactor (1/4″ O.D.), then heated to 700 °C for three hours under a hydrogen flow of 20 mL/min at one atmosphere. Then, at 700 °C, the hydrogen flow was stopped and replaced by $N_2$ at a flow of 30 mL/min. This continued for a period (typically 60 min) until no hydrogen was detected from the line using an online Gas Chromatograph (GC). The GC was equipped with a Thermal Conductivity Detector (TCD) to monitor molecular hydrogen. Hydrogen was analyzed with a Haysep Q packed column (2 m long, 1/8″ O.D.) at 45 °C connected to the TCD by using $N_2$ as a carrier gas. Once the complete reactor system was under pure $N_2$, the reaction was started by adding 3 vol. % of steam into the $N_2$ flow, which was sent in a continuous mode to the reactor. Pulses (of 0.5 mL each) from the reactor outlet were sent to the GC via a six-way valve during the reaction. The reaction continued until no hydrogen (from the water) was detected due to the complete oxidation of the oxide material. The reaction typically took 30 min.

All experimental work was conducted at the SABIC research centers of KAUST and Riyadh.

## 4. Conclusions

In this work, the extent of reduction by Ar ion sputtering of metal cations in $CeO_2$, $Ce_{0.95}Fe_{0.05}O_{2-\delta}$ and $Ce_{0.75}Fe_{0.25}O_{2-\delta}$ is studied by core and valence level spectroscopy. This is conducted to further study the reasons for the increased reaction rates of thermal water splitting when a fraction of Ce cations in $CeO_2$ are substituted with Fe cations. Ar ion sputtering resulted in an increase in the intensity of the Ce4f lines at about 1.5 eV below the Fermi level when compared to the O2p lines. Moreover, the XPS Ce5p/O2s ratio was found to be sensitive to the degree of reduction, which is attributed to a higher charge transfer from the oxygen to Ce ions upon reduction. This increase is concomitant with the increase in the XPS Ce3d spectra of the fraction of $Ce^{3+}$. Both increases are found to be higher for reduced $Ce_{0.95}Fe_{0.05}O_{2-\delta}$ when compared to $Ce_{0.75}Fe_{0.25}O_{2-\delta}$. On the other hand, the XPS Fe2p spectra showed no preferential increase between the two mixed oxides. Because water splitting to molecular hydrogen was found to be higher on $Ce_{0.95}Fe_{0.05}O_{2-\delta}$ than on $Ce_{0.75}Fe_{0.25}O_{2-\delta}$, it is postulated that the active sites for the reaction are those of $Ce^{3+}$ cations and not metallic Fe.

**Supplementary Materials:** The following supporting information can be downloaded at https://www.mdpi.com/article/10.3390/inorganics12020042/s1: Figure S1. Normalized XPS Ce5p and O2s spectra of as-prepared $CeO_2$ after 5 min of argon ion sputtering. Figure S2. Curve fitting of the XPS Ce5p and O2s peaks for the fresh samples A and C and argon-ion-sputtered samples B and D, $Ce_{0.95}Fe_{0.05}O_{2-\delta}$ and $Ce_{0.75}Fe_{0.25}O_{2-\delta}$, respectively.

**Funding:** This research received no external funding.

**Data Availability Statement:** Data can be requested directly from the author.

**Acknowledgments:** The author thanks Yahya Al-Salik (SABIC, STC-KAUST) for preparing the mixed oxides and for his technical help throughout the study and Toseef Ahmed (SABIC, STC-Riyadh) for the XPS data acquisition.

**Conflicts of Interest:** The author declares no conflicts of interest. SABIC was not involved in the study design, analysis, interpretation of data, the writing of this article or the decision to submit it for publication.

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
