# Peer review of "A Core and Valence-Level Spectroscopy Study of the Enhanced Reduction of CeO2 by Iron Substitution—Implications for the Thermal Water-Splitting Reaction"

_inorganics, doi:10.3390/inorganics12020042_

Round 1

Reviewer 1 Report

Comments and Suggestions for Authors

The manuscript presented by Hicham Idriss is dealt with the surface properties of two Fe-substituted Ce oxides and the blank CeO2 as well, characterized by XPS technique, which provided a supplementary study on the basis of the author's previous paper. This study, I think, is necessary and decent for the readers to more deeply understand the correlation of the thermal water splitting reaction with the surface properties of CeO2-based catalysts and the promoting effect of Fe. Here, three suggestions are provided for consideration as followed:

1) After a introduction of the previous paper (ref. 17), the author should give several sentences to express the purpose of the present paper;

2) The procedure of the performance evaluation of water splitting reaction in the Experimental Section should be presented. Moreover, the inconformity in the operating conditions of this study with the previous paper (ref. 17) should also be explained;

3) Adding the text involved the objective of Ar-ions sputtering is suggested.

Author Response

I thank the referee for her/his positive opinion about the manuscript and for providing valuable comments.

  • After a introduction of the previous paper (ref. 17), the author should give several sentences to express the purpose of the present paper;

Answer:

A paragraph is added to further explain the purpose of the manuscript.

  • The procedure of the performance evaluation of water splitting reaction in the Experimental Section should be presented. Moreover, the inconformity in the operating conditions of this study with the previous paper (ref. 17) should also be explained;

Answer:

The procedure of the water-splitting reaction in this study is now added to the experimental section.

  • Adding the text involved the objective of Ar-ions sputtering is suggested.

Answer:

A text explaining the reasons for using Ar ions to reduce the oxides is now added in the results section.

Reviewer 2 Report

Comments and Suggestions for Authors

The manuscript reports a very interesting study on Fe-doped ceria suitable as catalyst. I have some minor points to be clarified before publication:

-          - In the Introduction, Page 2, bullet 2 (“charge transfer”) it is reported: “substitution with reducible higher valence cations”; check if reducible is correct considering the subsequent sentence. Then, in such a sentence it is written “meal cation” but I suppose author intends “metal cation”.

-          - In the experimental part as source of Iron it is indicated Iron nitrate but the valency of Iron should be added as in Cerium Nitrate

-          - The chemical formula reported in the text are Ce0.95Fe0.05O2 and Ce0.75Fe0.25O2, but due to oxygen vacancies formed by Fe-doping the stoichiometric coefficient of O is less than 2

-         - It would be useful for the reader that the process with which hydrogen is produced (Figure 7) is indicated.

Author Response

I thank the referee for her/his positive opinion about the work and for providing further comments to improve the manuscript.

-          - In the Introduction, Page 2, bullet 2 (“charge transfer”) it is reported: “substitution with reducible higher valence cations”; check if reducible is correct considering the subsequent sentence. Then, in such a sentence it is written “meal cation” but I suppose author intends “metal cation”.

Answer:

Thank you very much.  Indeed, the word reducible is not right.  It has been replaced by the word oxidizable. The word meal was corrected by the word metal.

-          - In the experimental part as source of Iron it is indicated Iron nitrate but the valency of Iron should be added as in Cerium Nitrate

Answer:

The oxidation state of iron in iron (III) nitrate is added.

-          - The chemical formula reported in the text are Ce0.95Fe0.05O2 and Ce0.75Fe0.25O2, but due to oxygen vacancies formed by Fe-doping the stoichiometric coefficient of O is less than 2

Answer:

I agree with the referee.  For all chemical formulas, the stoichiometric coefficient of O is now adjusted to O(2-delta), where delta is less than 0.5.

-         - It would be useful for the reader that the process with which hydrogen is produced (Figure 7) is indicated.

Answer:

The procedure of the water-splitting reaction in this study is now added to the experimental section.